# Improving Catalytic Activity of “Janus” MoSSe Based on Surface Interface Regulation

**DOI:** 10.3390/molecules27186038

**Published:** 2022-09-16

**Authors:** Mingqian Wang, Xin Wang, Ming Zheng, Xin Zhou

**Affiliations:** 1Public Teaching Department, Heilongjiang Institute of Construction Technology, Harbin 150000, China; 2MIIT Key Laboratory of Critical Materials Technology for New Energy Conversion and Storage, School of Chemistry and Chemical Engineering, Harbin Institute of Technology, Harbin 150000, China

**Keywords:** Janus MoSSe, electrocatalysis, density functional theory, HER, OER

## Abstract

The monolayer Janus MoSSe is considered to be a promising catalytic material due to its unique asymmetric structure. In order to improve its catalytic performance for hydrogen evolution reactions (HERs) and oxygen evolution reactions (OERs), many attempts have been made. In this work, a series of transition metal (TM) atoms were anchored on the Janus MoSSe surface to screen effective TM single-atom catalysts for HERs and OERs through density functional theory (DFT) calculations. Fe@MoSSe presents excellent HERs performance and Ni@MoSSe presents excellent catalytic performance for OERs with extremely low over-potential of 0.32 V. The enhanced activity is attributed to the modest energy level of the d band center of the transition metal atom, and the transition metal atom improves the conductivity of the original MoSSe and offers unoccupied states near the Fermi level. At the same time, the anchoring of transition metal atoms redistributes the charge in the MoSSe system, which effectively regulates the electronic structure of the material itself. The strain calculation shows that the activity of the catalyst can be improved by reasonably adjusting the value of the applied strain.

## 1. Introduction

With the development of the economy and society, people are gradually realizing that excessive use of fossil energy will bring about a series of environmental problems such as the greenhouse effect, and therefore the development of efficient renewable energy is the only way to achieve the sustainable development of human society [1,2,3,4]. With continuous efforts in recent decades, various renewable energy sources have been developed and utilized. Electrocatalytic water splitting has been identified as a promising approach for renewable conversion and storage technologies, which involve the hydrogen evolution reaction (HER) and oxygen evolution reaction (OER) [5,6,7]. Electrocatalysis can effectively utilize intermittent energy sources, such as solar, wind and tidal power generation. On the other hand, the reaction efficiency of HERs and OERs also determines the performance and applicability of fuel cells. The fuel cells have the advantages of high energy efficiency and low environmental pollution, and are considered as ideal devices for converting chemical energy into clean energy [8]. Water splitting provides fuel for fuel cells and provides a promising and environmentally friendly way to convert and store clean renewable energy. Currently, Pt- and Ir-based materials are the best catalysts for the HER and OER, respectively. However, their practical application is severely limited by the high price and low abundance [9,10,11]. Searching for alternative electrocatalysts with low costs, high stability and excellent activity is necessary.

Among the potential catalytic materials currently developed, two-dimensional (2D) transition-metal dichalcogenides (TMDs) have emerged as promising substitutes for precious metal electrocatalysts because of their larger reaction area and unique electronic properties [12,13,14,15,16,17]. However, the catalytic performance of TMDs for HERs and OERs is still relatively poor compared to Pt- and Ir-based materials due to the limited highly active edge sites [18,19,20,21]. Recently, two independent research groups reported two different synthesis strategies for a new class of materials, “Janus” MoSSe [22,23], both of which synthesized highly asymmetric Janus monolayers and provided a structure with an intrinsic strain and electric field within the monolayer [24]. MoSSe is a special Janus monolayer whose structure is similar to the combination of MoS_2_ and MoSe_2_. Due to the special asymmetrical structure, it is expected to have better catalytic activity than pure TMD materials [25]. The intrinsic strain and intrinsic electric field of “Janus” MoSSe monolayers allow this new material to be applied on a large scale in fields such as photocatalysis, gas sensing, and electrocatalysis.

Li et al. reported that the Mo atom on the single-layer “Janus” MoSSe containing an S defect has good catalytic performance for the nitrogen reduction reaction (NRR) [26]. Shi et al. performed theoretical calculations to study the effect of intrinsic defects on the HER of Janus MoSSe monolayers, and found that point defects, including S/Se/Mo vacancies, were better than that of pristine MoSSe monolayers for HER performance, but it is still inferior to platinum [27]. Furthermore, related DFT studies have shown that the basal planes of “Janus” TMD materials can be activated without the need for large applied tensile strains and without significant density vacancies [28]. Ma et al. studied the possibility of MoSSe as a highly efficient photocatalyst for water splitting, taking into account the effects of isotropy and uniaxial strain [29]. In addition, Zhao et al. also explored the possibility of Co-doped monolayer MoSSe as a photocatalyst for water decomposition by first-principles calculations [30].

Single-atom catalysts (SACs) combine the high activity of homogeneous catalysts with the stability of heterogeneous catalysts and have great potential in various reactions, including HERs [31]. It is important to choose a suitable substrate for SACs, which can effectively immobilize individual atoms to prevent their agglomeration, and at the same time can provide abundant electrons. As a special ternary transition metal chalcogenide, the “Janus” MoSSe has a band gap and interlayer spacing between MoS_2_ and MoSe_2_, and its adsorption of water molecules is stronger than that of MoS_2_ and MoSe_2_. The destruction of structural symmetry results in the redistribution of electrons between the S atom and the Se atom on the surface, and efficiently accumulates the charge onto the metal atoms on the corresponding side of the Janus monolayer. Therefore, anchoring suitable transition metals on the “Janus” MoSSe surface is a candidate material for improving water splitting, and will hopefully achieve the goal of designing multifunctional water splitting catalysts.

In this work, we performed DFT calculation-screening studies by anchoring different single transition metal (TM) atoms on “Janus” MoSSe monolayers to explore the feasibility of single-atom anchoring of “Janus” MoSSe monolayers (TM@MoSSe) as electrocatalysts for water splitting. In addition, how the electronic structure and catalytic activity of “Janus” MoSSe depend on strain engineering is still unknown. Therefore, after anchoring a single atom on “Janus” MoSSe, the effect of applied strain on the catalytic activity was further considered and its nature was further studied.

## 2. Computational Methods 

All the structure optimization calculations were utilized by the Vienna Ab-initio Simulation Package (VASP) [32,33], and the spin polarization was considered. Projector Augmented Wave (PAW) pseudo-potential substitutes for the interaction of core electrons [34,35], and the generalized gradient approximation (GGA) in the Perdew–Burke–Ernzerhof (PBE) form, were adopted to describe exchange–correlation functional effects [36]. The van der Waals (vdW) interactions were considered the DFT-D2 method of Grimme [37]. Plane-wave basis functions were expanded with cut-off energy of 400 eV and k-points were sampled using 3 × 3 × 1 Monkhorst-Pack mesh for the structural relaxations. Computations were continued until the energy and force converged within 1 × 10^−5^ eV and 0.03 eV/Å, respectively. In order to avoid the interaction of the periodic units, the thickness of the vacuum layer was set to 18 Å. To confirm the origin of the catalytic activity of TM@MoSSe, density of states (DOS) calculations and Bader charge analysis were performed. The 7 × 7 × 1 Monkhorst–Pack grid was used for the DOS calculation, and the other parameters were consistent with the above.

The stability of the SACs can be verified from the adsorption energy of various TMs anchored on MoSSe and the dissolution potential (*U*_diss_, vs. SHE); both ensure the stability of the TM@MoSSe under an electrochemical environment [29,30].
(1)$$\Delta E_{\mathrm{TM}-\mathrm{ads}}=E_{\mathrm{TM}-MoSSe}-E_{MoSSe}-E_{\mathrm{TM}}$$
where Δ*E_TM–MoSSe_*, represent the total energy of the TM atoms anchored on the MoSSe, *E_MoSSe_* and *E*_TM_ represent pure MoSSe, and single TM atom (using the energy of a single TM in the gas phase), respectively. Clearly, the more negative the adsorption energy, the higher the thermodynamic stability.
(2)$$U_{\mathrm{diss}\;}=U_{\mathrm{diss}\;}^{0}(\mathrm{metal}\;,\;\mathrm{bulk})-\frac{Ea}{N_{e}|e|}$$
*U*^0^_diss_ (metal, bulk) represents the standard dissolution potentials (pH = 0) of pure bulk metal, and Ne is the number of electrons in dissolution. A more positive *U*_diss_ indicates that the TM atoms are electrochemically more stable against dissolution.

To further verify the stability of the screened TM@MoSSe, the thermodynamic stability of these systems was analyzed with ab initio molecular dynamics (AIMD) simulations. At the start of AIMD simulations, the TM@MoSSe samples were assigned an initial temperature of 100 K, and then heated up to the desired temperature of 500 K over 3 ps. The canonical ensemble (NVT) with Nosé thermostat was adopted at 500 K for the total time of 10 ps with 1 fs time step. Additionally, all models in this work were constructed and post-processed using the Device Studio program. The Device Studio program provides a number of functions for performing visualization, modeling and simulation. The AIMD simulation Bader charge analysis using DS-PAW [34] software is also integrated in the Device Studio program [38].

## 3. Results and Discussion

### 3.1. Geometric Structure and Stability of TM@MoSSe

The structures of adsorbed TM single atoms on MoSSe are shown in Figure 1a,b. Due to the special asymmetric structure of two-dimensional Janus material MoSSe, comprehensive consideration must be taken into account when considering the adsorption sites of transition metal single atoms on the surface of the material, and then stability analysis and performance research. Three different adsorption sites were considered for the S atomic layer on MoSSe, namely the Hollow site, the S atomic site, and the Mo atomic site. Similarly, on the Se atomic layer side, three different adsorption sites of transition metals on MoSSe were also considered, namely the Hollow site, Se atomic site, and Mo atomic site.

In order to find the most suitable SACs for OERs and HERs, the stability of different TM atom loads such as Fe, Co, Ni, Ru, Rh, Pd, Ag, Ir, Pt and Au on MoSSe surface was investigated. Firstly, the adsorption energies of different TM atoms on MoSSe were calculated to evaluate the stability and determine the geometric structures of different SACs. As shown in Figure 1c,d, the adsorption energy calculation results show that Co, Ni, Ru, Rh, Pd, Ir, Pt, and Au had the most negative adsorption energy at the Mo site, followed by the Hollow site, and the most positive adsorption energy at the S/Se site on base MoSSe. The adsorption energy of a Ag single atom at Mo site and Hollow site was similar, and the adsorption energy value at the S/Se site was the most positive and the adsorption was the most unstable. Compared with the adsorption of other TM single atoms on MoSSe, the adsorption of Ag single atoms was weak (the adsorption energy was −0.57~−0.91 eV), and Ag single atoms cannot form stable single-atom catalysts on the surface from an energy point of view. In conclusion, TM (Fe, Co, Ni, Ru, Rh, Pd, Ir, Pt, and Au) single atoms can stably adsorb on Mo atomic sites of both the Se side and S side on MoSSe, and further comparison of adsorption energies showed that the adsorption of TM single atoms on the S side of MoSSe was more stable. Therefore, the stable model of TM@MoSSe was preliminarily determined as TM single atoms supported on Mo sites in the S layer on MoSSe.

It is worth noting that the dissolution potential (U_diss_) is also very important for the stability evaluation of electrocatalytic materials [39]. More negative adsorption energy means that TM atoms adsorbed on MoSSe are thermodynamically stable, while the corrected U_diss_ indicates that TM atoms are more electrochemically stable. The stability of the material in the electrochemical reaction environment can be indicated only when both of them reach a certain standard. As shown in Figure 1e,f, the calculation results of the dissolution potential show that Fe Co, Ni, Ru, Rh, Pd, Ir, Pt, and Au had the largest dissolution potential at the Mo site on the MoSSe, followed by the Hollow site, and the smallest at the S/Se site. The dissolution potential of Fe and Ag at the Mo site and Hollow site was similar, and the solubility potential at the S/Se site was the least, which means that it was the most unstable. The calculation result of dissolution potential is consistent with the adsorption energy, that is, the stability model of TM@MoSSe is TM (Fe, Co, Ni, Ru, Rh, Pd, Ir, Pt, Au) loaded on the Mo atomic site in the S layer of MoSSe.

Thermodynamic calculation alone is not enough to describe the stability of materials at practical application temperature. On the basis of energy calculation, the stability of materials is further determined by AIMD calculation. Firstly, the TM@MoSSe (TM = Fe, Co, Ni, Ru, Rh, Pd, Ir, Pt, Au) samples were assigned an initial temperature of 100 K, and then heated up to the desired temperature 500 K over 3 Ps. During the simulation process, the position of Au atoms on the Au@MoSSe surface changed with temperature, indicating that Au atoms cannot be anchored on the surface of MoSSe to form single atomic catalysts, as shown in Appendix A. For other TM@MoSSe (TM = Fe, Co, Ni, Ru, Rh, Pd, Ir, Pt) materials, the structure of a single atom anchored on MoSSe surface was very stable from 100 to 500 K.

In addition, we further performed an AIMD simulation for 10 ps. As shown in Figure 2, the total energy and the temperature oscillated near the equilibrium state. Meanwhile, the structure of the single atom anchored on MoSSe surface still had no significant change (the TM@MoSSe trajectories of AIMD simulation shown in Appendix A), indicating the high stability of the TM@MoSSe material. Therefore, we can conclude that the single TM (Fe, Co, Ni, Ru, Rh, Pd, Ir, Pt) atom anchored on MoSSe surface is stable and the next discussion of catalytic performance is based on the screening in this section.

### 3.2. HER and OER Catalytic Activity

The original unmodified surface of MoSSe is inert and not an excellent catalyst for HERs and OERs. After screening the stable TM@MoSSe structure, the single atom anchored on the surface was used as the catalytic active site to explore their HER and OER performance. To evaluate the HER activity of TM@MoSSe, we calculated the Gibbs free energy (ΔG_H_) of the hydrogenation process based on the method proposed by Norskov et al. [40,41], in which the free energy of H^+^+ e was replaced by half the chemical potential of a hydrogen molecule. The absolute value of ΔG_H_ signifies the energy barrier to be overcome for HERs, a positive value of ΔG_H_ indicating that the binding strength between the H atom and catalyst is strong and that it is difficult to desorb, while a negative value of ΔG_H_ indicating that the interaction between H and the catalyst is weak and that it is difficult to adsorb.

As shown in Figure 3c, all TM@MoSSe catalysts exhibited better HER activity than the original MoSSe materials, indicating that anchoring a single transition metal atom on MoSSe is a feasible method to improve the catalytic performance of MoSSe materials. It is worth noting that the values of Fe@MoSSe, Ru@MoSSe, Rh@MoSSe, and Pt@MoSSe ΔG_H_ were 0.08 eV, −0.05 eV, 0.03 eV, and −0.06 eV, which are close to 0. As is well-known, the high cost and low storage of these precious metals (Ru, Rh, Pt) seriously limit their practical application in energy conversion and storage. The Fe element, with low cost and high reserves, is one of the candidate materials for catalysts that are expected to replace precious metals. Furthermore, it is reported that a variety of Fe single-atom catalysts have been successfully synthesized and used for HER electrocatalysts, which can ensure good stability and activity in acidic [42] or alkaline [43] electrolyte media. Therefore, the Fe@MoSSe can be used as a potential HER electrocatalytic material.

The OER performance of the original MoSSe was calculated by considering two different reactive active sites (i.e., the Mo atom site on the S side and the Mo atom site on the Se side). As shown in Figure 4a, the calculated Gibbs free energy of the four-step OER reaction process showed that the OER overpotential was 1.96 V at the Mo atom site on the S side and 1.47 V at the Mo atom site on the Se side. In the third step, the energy required for the reaction of O* with OH* to generate the intermediate of HOO* was the largest. This step is a key step to determine the overpotential of the electrocatalytic oxygen evolution reaction on MoSSe. The large overpotential indicated that the catalysis activity of MoSSe needs to be improved, and MoSSe without any surface modification is not an excellent electrocatalytic material that can replace precious metals.

The OER reaction process includes four elementary steps, each of which is accompanied by proton and electron transfer. For the elementary step, the change of Gibbs free energy (ΔG) of the four step reactions can be calculated according to the first-principles calculation. The ΔG value of the largest of the four-steps will determine the overpotential of OER (η). Figure 3a shows the OER overpotential calculation results of TM@MoSSe (TM = Fe, Co, Ni, Ru, Rh, Pd, Ir, Pt) from left to right. The overpotential was 1.02 V, 0.79 V, 0.32 V, 1.16 V, 0.68 V, 0.55 V, 0.93 V and 0.55 V, respectively. It is worth noting that Ni@MoSSe had the smallest overpotential value (η = 0.32 V) during a series of TM@MoSSe, and the dissociation of OH* to O* is the key step to determining the OER overpotential, shown in Figure 4b. The overpotential of Ni@MoSSe was 1.15 eV lower than that of the original MoSSe, and Ni@MoSSe has better catalytic performance than the typical OER catalyst IrO_2_ (η = 0.42 V). In previous experimental studies [44], IrOx was found to be the only stable catalyst in acidic media compared with non-precious metal catalysts. In addition, although IrOx is unstable, no non-precious metal catalysts had the same activity as IrOx under the same conditions. In this work, the calculation results show that it is possible to break this relationship by preparing non-precious metal single-atom catalysts. The dissolution potential and AIMD simulation indicated that the non-noble metal single-atom anchored MoSSe catalyst has strong stability. As is well-known, single-atom catalysts improve the utilization of metals and increase the number of active sites. By anchoring different metals on the surface of MoSSe, the d-band center of the catalyst is further adjusted to reduce the overpotential of OERs.

All free energy diagrams for the OER of the TM@MoSSe at an electrode potential of U = 0 V are shown in Appendix A.

Interestingly, as shown in Appendix A, when the Gibbs free energy of TM@MoSSe changes, it is found that the adsorption of Gibbs free energy of intermediate O* is significantly correlated with the number of d electrons adsorbed on transition metal atoms. In the same period, as the number of transition metal d electrons increases, the ΔG value of the second step reaction also increases. The above phenomena indicate that the number of d electrons of TM atoms will affect the energy change of the OER process, which can provide some guidance for the follow-up study on the source of OER activity of TM@MoSSe. In addition, under the experimental conditions, some single-atom catalysts formed clusters on the surface, thus affecting the catalytic activity of the catalysts. Therefore, a Ni4 cluster model was established on the MoSSe surface to investigate the effect of clusters on the catalytic activity. Furthermore, the results show that Ni@MoSSe can maintain good OER and HER activities even if isolated Ni atoms are clustered into small clusters, as shown in Figure 4c,d. See the Appendix A for a detailed discussion.

### 3.3. The Origin of Catalytic Activity of TM@MoSSe

By anchoring transition metal single atoms on the surface, the catalytic performance of the original MoSSe was greatly improved. In order to understand the source of excellent catalytic activity, the charge of TM@MoSSe was calculated and analyzed. As shown in Table 1, the series TM@MoSSe Bader charge can be used to quantitatively calculate the charge of the system, and the number of electrons around the atom can be obtained, so as to approximate the valence of the atom. The data analysis shows that after the adsorption of transition metal single atoms, the charge of the material is transferred to different degrees, in which Se and S atoms obtain electrons, and the S atom obtains more charges than Se. The main atoms that lose electrons are Mo atoms and doped transition metals. This shows that after the transition metal single atom is anchored, the charge in the MoSSe system is redistributed, the electrons in the Se layer are transferred to the S layer and the transition metal atoms on the surface.. The change of the electronic structure is the main reason for the difference in material performance, and the intervention of the transition metal single atom effectively regulates the electronic structure of the material itself.

The good conductivity of the catalyst itself can greatly eliminate the energy barrier at the catalyst–electrolyte interface and the electrode interface in the electrocatalytic reaction, and ensure high-energy conversion efficiency. The density of states of the original MoSSe was calculated. As shown in Appendix A, the original MoSSe is a semiconductor material with a band gap of 1.62 eV. From the density-of-states diagram analysis, it can be seen that, compared with non-metallic elements, such as S and Se, the d-electronic states of Mo atoms contribute more to the electronic states near the Fermi level.

The OER performance of Ni@MoSSe is very prominent in a series of TM@MoSSe. In order to explore the essential reason for the catalytic activity of Ni@MoSSe, the DOS was analyzed. As shown in Appendix A, the band gap of the material decreased to a certain extent after the single atom Ni was loaded on the surface of MoSSe, indicating that the conductivity of MoSSe was improved to a certain extent after the modification of single atom Ni. At the same time, the other TM@MoSSe also had the same tendency, indicating that transition metal loading can improve the catalytic activity of MoSSe material.

For metal catalysts, the active element composition, chemical state, size, and the crystal structure of all sorts of complicated factors, such as extremely subtle structure change, can bring huge performance change. Therefore, it is very difficult to adjust the performance of the metal catalyst. According to the d-band center theory, the d-band center of a series of TM@MoSSe metals was explored and fitted with overpotential. It was found that there was a good linear relationship between the d-band center and material OER overpotential (R^2^ = 0.85), as shown in Figure 5. In the TM@MoSSe system, the stability and activity of the catalysts are improved and the OER overpotential of the materials is decreased with an increase of the binding strength between the transition metals and the MoSSe surface.

### 3.4. The Influence of Strain Effect

Strain engineering is an effective strategy to adjust the surface catalytic activity and has been widely used in catalyst design regions. Compared with bulk materials, two-dimensional materials with certain surface strain exhibit different catalytic properties due to their different lattice constants and electronic structures. The performance calculation and screening results of the TM@MoSSe showed that the Ni@MoSSe catalyst has excellent OER performance and Fe@MoSSe has better HER activity. In order to deeply understand the effect of surface strain on the surface structure and properties of MoSSe, Fe@MoSSe and Ni@MoSSe catalyst materials were selected to study the strain effect in combination with the performance and the economic cost of practical application.

#### 3.4.1. Structure and Stability

In order to simulate the strain of two-dimensional materials, as shown in Appendix A, the lattice length was uniformly reduced or increased in directions a and b of the structural model without changing the length in the direction c, so that the MoSSe surface generated a strain in the range of −10% (negative value indicates compressive strain) to 10% (positive value indicates tensile strain) in the directions a and b. Surface strain is defined as strain = ((I − I_0_)/I_0_) × 100%, where I is the lattice parameter of the strained surface and I_0_ is the lattice length of the unstrained surface.

After the external strain is applied on the surface of the catalyst, the stability of the surface is very important for the subsequent electrocatalysis application. Here, adsorption energy was used to investigate the effect of strain on Ni@MoSSe and Fe@MoSSe stability. As shown in Appendix A, the adsorption energies of the Ni@MoSSe and Fe@MoSSe were below −2.2 eV, which indicates that the stability of the two materials was well under the action of external strain. With the increase of tensile strain, the adsorption energies of Ni@MoSSe and Fe@MoSSe showed a decreasing trend. In the range of −10~10%, the greater the applied strain, the higher the stability. The sensitivity of Fe@MoSSe surface stability to strain is greater than that of Ni@MoSSe, indicating that the effect of strain on surface stability is related to the crystal surface structure. The variation of structural parameters of the two structures with surface strain was further investigated. As shown in Appendix A, as the tensile strain increased, the distance between the S atom layer and the Se atom layer increased, and the distance between Mo atoms in the same layer decreased. As shown in Appendix A, the compressive strain made the electronic state distribution on the surface of Ni@MoSSe and Fe@MoSSe more compact, while the tensile strain makes the state density distribution more extensive. This is because the tensile strain reduces the orbital overlap between atoms and makes electrons more localized, while the compressive strain has the opposite effect. The above changes in electronic structure can be explained by the changes in crystal surface structure. The stretching of the surface will reduce the overlap of atomic orbits, thus narrowing the distribution of electronic density of states; the surface compressive strain will increase the overlap of atomic orbitals in the system, and thus the electronic state distribution will be broadened. Therefore, the application of strain on the surface may cause changes in the adsorption energy of surface species, thus regulating its electrocatalytic performance.

#### 3.4.2. OER and HER Performance of Materials under Strain Effect

As shown in Figure 3b, the results of the OER overpotential calculation for Ni@MoSSe under different stresses show that when the stress was −6%, the overpotential was 0.35 eV, and the OER performance of the material was close to that without additional strain. When the stress was −10%, −6%, 0%, 2%, 6%, and 10%, the OER overpotential increases significantly compared with the absence of external strain; that is, to adjust the OER overpotential of Ni@MoSSe by changing the stress, the external strain needs to be adjusted to a reasonable value. Figure 3d shows the results of HER overpotential calculation for Fe@MoSSe under different strains. When the strains were −10%, −6%, −2%, and ΔGH* was −0.04 eV, −0.03 eV and 0.03 eV, respectively, the HER performance was excellent, which can improve the electrocatalytic hydrogenation performance of Fe@MoSSe through external compression stress.

#### 3.4.3. Mechanism Study

The variation of crystal surface structure can explain the change in electronic structure, so the change in the d-band center can be related to the change of spacing between the S and Se layers. As shown in Figure 6a, the spacing between the S and Se layers decreased with increasing strain, while the distance between the Mo atoms in the same layer increased with increasing strain in Ni@MoSSe. Furthermore, the d-band center shows an increasing trend in the strain range of −10~10%. For Fe@MoSSe, the atomic layer spacing between the S layer and Se layer decreased with increasing strain, while the distance between Mo atoms in the same layer increased with increasing strain. In the strain range of 0~10%, the d-band center of the band showed an increasing trend, as shown in Figure 6b. However, within the compressive strain range, the change of the center of the d-band was not regular, which also resulted in the abnormal adsorption energy of the adsorbed intermediates during the reaction. This indicates that the simple d-band center theory is no longer sufficient to fully explain the surface properties with strain. Therefore, the centers of each d-orbit need to be calculated separately to gain a better understanding of the catalytic behavior under different stresses.

As shown in Appendix A, within the strain range, the center of the d-band increased with the increase of external stress, and the changing trend of different d-orbital components was consistent, whereas the center moved up of d_x_^2^, d_xy_, and d_z_^2^, resulting in the center of the d-band being close to the Fermi level. Within the range of compression strain, the change of the d-band center was caused by the interaction of d_x_^2^, d_xy_, and d_z_^2^ orbits. The energy of d_x_^2^ and d_xy_ orbits is higher than that of d_z_^2^ orbits, which leads to the irregular change of the d-band center.

## 4. Conclusions

In summary, various TM@MoSSe single-atom catalysts have been constructed and screened by first-principles calculations. The catalytic performance and source of activity of HERs and OERs were studied, and the effect of strain on catalytic activity was further explored. The calculation results of adsorption energy and dissolution potential show that the Mo atom site on the S side of MoSSe is the most stable site for the adsorption of transition metals. By AIMD simulation, six stable TM@MoSSe single atom catalysts were further screened out. It was also shown that simple energy calculation is not enough to describe the stability of materials at practical application temperatures.

The result of the Bader charge suggesting an anchoring of transition metal atoms redistributes the charge in the MoSSe system, which effectively regulates the electronic structure of the material itself. By calculating and comparing the OER and HER overpotentials, it was found that the energy barrier of the OER reaction can be greatly reduced by Ni anchoring on the origin MoSSe surface, and non-precious metal Fe@MoSSe can be used as HER electrocatalysts with practical application value. Therefore, Fe@MoSSe and Ni@MoSSe were selected to investigate the influence of external strain on the performance of the catalyst. The analysis shows that stretching or compression changes the overlap of atoms on the surface of two-dimensional materials, and the interaction between d_x_^2^, d_xy_, and d_z_^2^ plays an important role in the change of the position of the d-band center in the material system.

## Figures and Tables

**Figure 1 molecules-27-06038-f001:**
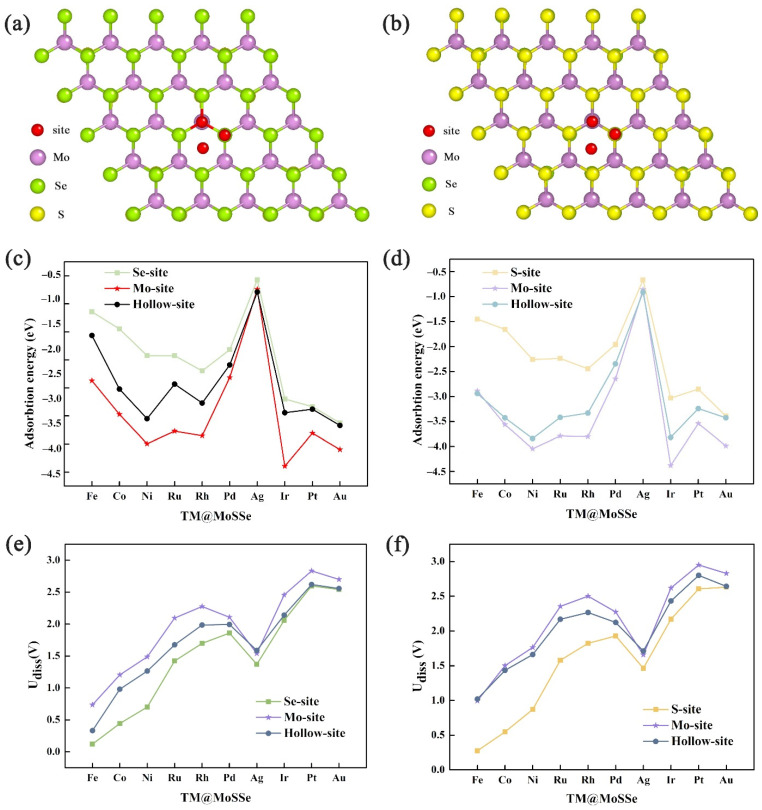
Adsorption structure of TM atoms anchored on (**a**) the S side of MoSSe and (**b**) the Se side of MoSSe, where purple, yellow, green and red balls represent the Mo, S, Se and TM. Adsorption energies of different TM atoms on (**c**) the Se side of MoSSe and (**d**) the S side of MoSSe. Dissolution potentials of different TM atoms on (**e**) the Se side of MoSSe and (**f**) the S side of MoSSe.

**Figure 2 molecules-27-06038-f002:**
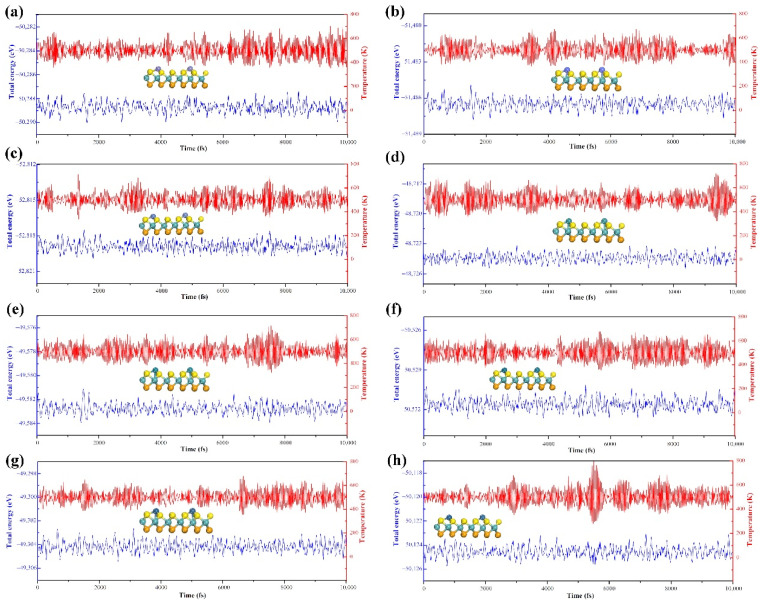
Variations of temperature and energy of against the time for the AIMD simulations of (**a**) Fe, (**b**) Co, (**c**) Ni, (**d**) Ru, (**e**) Rh, (**f**) Pd, (**g**) Ir, and (**h**) Pt@MoSSe. The simulation was run under 500 K for 10 ps with a time step of 1 fs.

**Figure 3 molecules-27-06038-f003:**
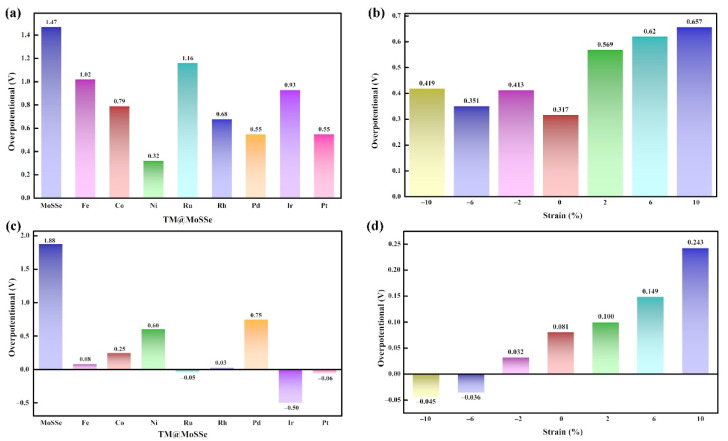
The OER overpotential of (**a**) TM@MoSSe and (**b**) Ni@MoSSe under different strain; the HER overpotential of (**c**) TM@MoSSe and (**d**) Fe@MoSSe under different strain.

**Figure 4 molecules-27-06038-f004:**
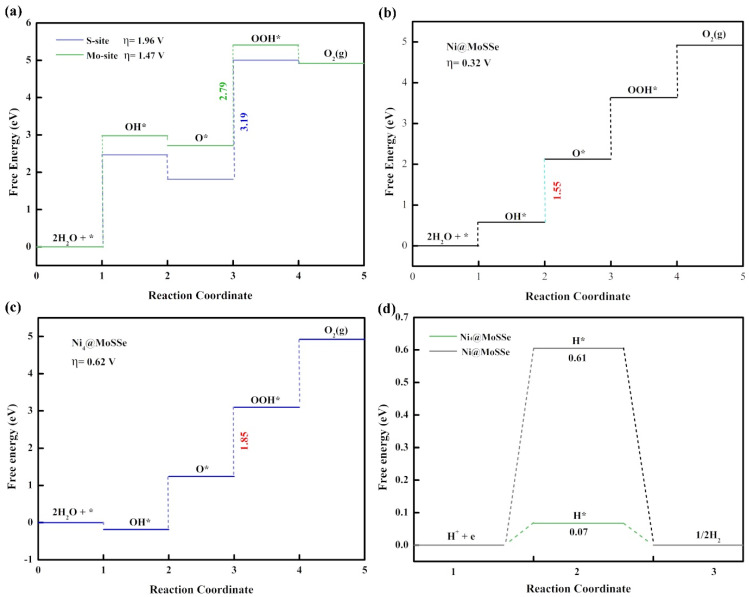
The free energy diagram for the OER of (**a**) pristine MoSSe, (**b**) Ni@MoSSe, (**c**) Ni_4_@MoSSe, (**d**) The free energy diagram for HER of the Ni_4_@MoSSe.

**Figure 5 molecules-27-06038-f005:**
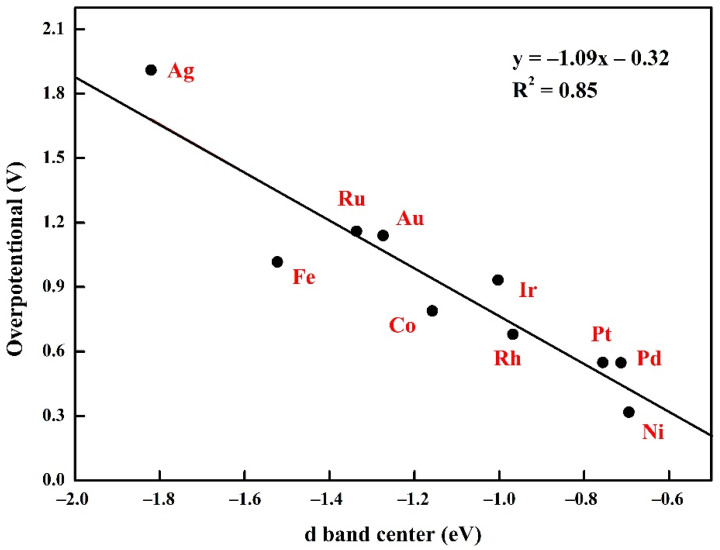
Calculated overpotential on different TM@MoSSe catalysts as a function of the corresponding d band center.

**Figure 6 molecules-27-06038-f006:**
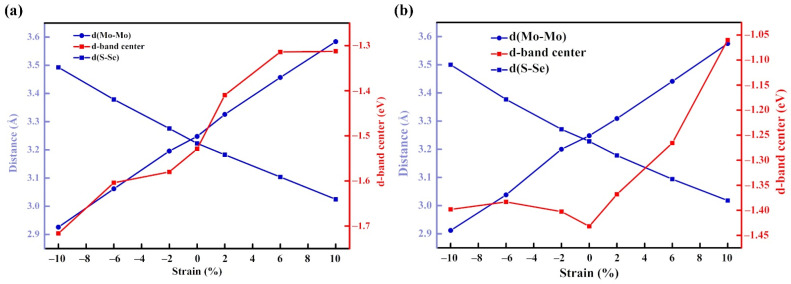
Effect of changing strain on structure and d-band center of (**a**) Ni@MoSSe, (**b**) Fe@MoSSe.

**Table 1 molecules-27-06038-t001:** Bader charge of TM@MoSSe. The positive values indicate the loss of electrons.

TM@MoSSe	q (Mo)	q (S)	q (Se)	q (TM)
Fe	1.063	−0.670	−0.462	0.609
Co	1.074	−0.662	−0.463	0.466
Ni	1.077	−0.654	−0.464	0.379
Ru	1.078	−0.651	−0.467	0.368
Rh	1.076	−0.636	−0.466	0.227
Pd	1.080	−0.637	−0.466	0.206
Ir	1.065	−0.620	−0.461	0.144
Pt	1.068	−0.614	−0.460	0.044

## Data Availability

Not applicable.

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
