# Peer review of "Improving Catalytic Activity of “Janus” MoSSe Based on Surface Interface Regulation"

_molecules, 2022, doi:10.3390/molecules27186038_

Round 1
Reviewer 1 Report
A separate document has been created for comments.

Reviewer 2 Report
The work by Zhou et al. reported that a series of monolayer Janus MoSSe anchored to different transition metal single-atom as a promising class of electrocatalysts for HER and OER by using DFT calculations. They found that Fe@MoSSe presents excellent HER performance and Ni@MoSSe presents excellent catalytic performance for OER with extremely low overpotentials. The subject is very interesting and important because it can contribute to the understanding of the catalytic mechanism of HER and OER and the design of new catalysts. The manuscript is well organized and prepared. So I recommend publication after addressing following issues.
1. The stability of single atoms loading on MoSSe surface was investigated by AIMD simulation. Do single atoms aggregate on their surfaces? How does the author handle the issue?
2. According to the calculated results of overpotential, Fe@MoSSe is not the best catalyst for OER, and the authors should elaborate on the reasons for choosing Fe @MoSSe for further study in the manuscript.
3. The latest advances in MoSSe catalysis should be provided in the introduction, especially in HER and OER.
4. Page 10, Section 3.4.1. what are the criteria for selecting strain ranges when considering strain effects? Will it cause the crystal form change ?
5. Some minor errors should be corrected:
Page 6, the pixel of Figure 2 should be improved to make them clearer;
Page 12, 'FER' should be changed to 'HER' in Figure 7 (b);
Page 11, “ When the stress is -10%, -2%, -2%, 2%, 6%, and 10%” should be changed to “-10%, -6%, , -2%, 0, 2%, 6%, and 10%”
